# A Novel OpenBCI Framework for EEG-Based Neurophysiological Experiments

**DOI:** 10.3390/s23073763

**Published:** 2023-04-06

**Authors:** Yeison Nolberto Cardona-Álvarez, Andrés Marino Álvarez-Meza, David Augusto Cárdenas-Peña, Germán Albeiro Castaño-Duque, German Castellanos-Dominguez

**Affiliations:** 1Signal Processing and Recognition Group, Universidad Nacional de Colombia, Manizales 170003, Colombia; 2Automatics Research Group, Universidad Tecnológica de Pereria, Pereira 660003, Colombia; 3Cultura de la Calidad en la Educación Research Group, Universidad Nacional de Colombia, Manizales 170003, Colombia

**Keywords:** brain computer interfaces, OpenBCI, EEG, drivers, distributed systems, neurophysiological

## Abstract

An Open Brain–Computer Interface (OpenBCI) provides unparalleled freedom and flexibility through open-source hardware and firmware at a low-cost implementation. It exploits robust hardware platforms and powerful software development kits to create customized drivers with advanced capabilities. Still, several restrictions may significantly reduce the performance of OpenBCI. These limitations include the need for more effective communication between computers and peripheral devices and more flexibility for fast settings under specific protocols for neurophysiological data. This paper describes a flexible and scalable OpenBCI framework for electroencephalographic (EEG) data experiments using the Cyton acquisition board with updated drivers to maximize the hardware benefits of ADS1299 platforms. The framework handles distributed computing tasks and supports multiple sampling rates, communication protocols, free electrode placement, and single marker synchronization. As a result, the OpenBCI system delivers real-time feedback and controlled execution of EEG-based clinical protocols for implementing the steps of neural recording, decoding, stimulation, and real-time analysis. In addition, the system incorporates automatic background configuration and user-friendly widgets for stimuli delivery. Motor imagery tests the closed-loop BCI designed to enable real-time streaming within the required latency and jitter ranges. Therefore, the presented framework offers a promising solution for tailored neurophysiological data processing.

## 1. Introduction

Using a Brain–Computer Interface (BCI), devices can be controlled by stimulating brain electrical activity with a wide range of applications, including neuromarketing and neuroeconomics [1,2], games and entertainment [3,4], security [5], frameworks for operating medical protocols like cognitive state analysis, rehabilitation of individuals with motor disabilities [6], diagnosis of mental disorders and emotion-based analysis [7], among others. A recent development in using BCI technology in educational contexts is also reported [8]. It encompasses a robust set of competencies critical for individuals to contribute actively to human development as part of the UNESCO Media and Information Literacy (MIL) methodology, as discussed in [9,10].

Despite the availability of numerous technologies for neurophysiological data acquisition in BCI, Electroencephalography (EEG) is the most common method for extracting relevant information from brain activity due to its high temporal resolution and low cost, portability, and low risk to the user [11]. Yet, scalp electrodes have severe disadvantages, such as non-stationarity, low signal-to-noise ratio, and poor spatial resolution. Moreover, adequate user skills are required to implement clinical protocols so that BCI must be involved under controlled laboratory conditions like in Motor Imagery (MI) [12]. As a result, signal acquisition, instrumentation, and software development procedures must be integrated effectively to validate clinical protocols for brain neural responses. The implementation of BCI frameworks with acceptable reliability has therefore become a challenging task.

Table 1 provides a comprehensive overview of the critical features supplied by leading BCI acquisition systems. In particular, a couple of BCI systems for MI are presented: Emotive EPOC+ (https://www.emotiv.com/epoc/, accessed on 1 October 2022) and B-Alert x10 (https://www.advancedbrainmonitoring.com/products/b-alert-x10, accessed on 1 October 2022). As a general rule, BCI designs should meet the following requirements when used for clinical purposes [13,14,15]: (i) Accounting for the high-quality acquisition of EEG data at moderate costs; (ii) Adaptable to various experimental settings, allowing for a wide range of processing complexity and scalability of applications, and (iii) Ability to accommodate specialized software to more general-purpose protocols. As regards the first case, a BCI is typically designed to perform uncomplicated tasks and thus conceived in a simplified configuration, including reduced state representations, low-frequency data transmission, scalp placements with a few electrodes, or data processing modules with a decreased computational burden. Nevertheless, several neurophysiological processes (such as concentration, alertness, stress, and levels of pleasure) can demand more versatility and enhanced technical requirements from BCI to gauge broader aspects of brain activity. In addition, clinical devices still need real-time data flow access since they are often dedicated to online analysis [16].

In the subsequent data processing analysis, three procedures are employed: signal preprocessing, feature extraction, and classification/prediction inference. Additional modules for control flow between devices and graphical user interfaces must also be incorporated. In standard EEG clinical setups, all those components are executed simultaneously [17,18]. Furthermore, BCIs often do not run on real-time operating systems, meaning system resources impact each component. To cope with this shortcoming, high-performance processing units running multiple processes on non-real-time operating systems can be involved. As a rule, BCI complexity tends to reduce by allocating computationally intensive tasks across a distributed system, resulting in improved system reliability and enhanced performance [19]. However, the use of closed-loop BCI systems for the analysis of brain neural responses implies data processing with extended complexity that requires synchronization of the following components: acquisition, signals database/storage, feature processing (extraction and classification), visualization (temporal, spectral, and spatial), command generation for actuators, command database, and feedback acquisition. Moreover, protocols with Event-Related Potential require designs with higher precision for marker synchronization, demanding low and consistent latency. These requirements for enhanced stability and processing capacity become critical in research settings, where centralized systems are vulnerable to slowing down due to unexpected computational demands [20,21].

Next, Table 2 shows examples of the standard software tools for BCI that are autonomous and run independently. Solutions for special applications like psychology, neuroscience, or linguistics are also being developed through proprietary software. This situation tends to hinder system extensibility due to limitations in data transmission protocols and compatibility with limited hardware [22,23,24]. Besides, these proprietary solutions may come with additional costs, technical expertise, and open-source benefits that require high-level programming skills with limited support for data acquisition [25]. By contrast, incorporating design strategies with open-source components may raise technology acceptance, lower creation costs, support collaborative development and make BCI designs accessible to a broader class of developers. This freedom to access and modify hardware and firmware provides ample opportunities to create custom drivers with cutting-edge features of functionality and performance [26,27]. Nevertheless, only one of BCI’s options in Table 1 is open-source. Still, a context-specific design and suitable drivers are also required in enhancing the technical features of baseline BCI frameworks [28,29].

Here, we present a new OpenBCI framework designed to meet the requirements of a broader class of MI protocols, embracing three components: (i) acquisition drivers enhanced for OpenBCI that offer high-level features, such as distributed and asynchronous data acquisition, enabled through a Python module wrapping the vendor’s SDK; (ii) a distributed system strategy to eliminate latencies in BCI experiments by accurately synchronizing markers and allowing for simultaneous data acquisition; (iii) a framework interface to integrate a full-featured BCI into a single application, streamlining the deployment process and reducing potential failure points. The OpenBCI framework leads to the automatic creation of ready-to-use databases and simplifies testing and design procedures, enhancing repeatability and speeding up debugging. Automatic marker synchronization is also offered to improve the integration of various features from multiple systems into a single one. Results of experimental testing performed for the MI paradigm show effectiveness, including binary deserialization, data transmission integrity and latency, EEG electrode impedance measurement, and marker synchronization.

The remainder of this paper is organized as follows: Section 2 describes the methods and software tools integrated into our development. Section 3 and Section 4 depict the experimental setup and results regarding an illustrative example within a motor imagery paradigm based on our OpenBCI tool. Finally, Section 5 outlines the conclusions and future work.

## 2. Materials and Methods

The presented OpenBCI-based framework embraces three steps: (i) custom drivers developed to optimize system performance; (ii) driver integration into distributed systems to increase accessibility and usability; (iii) high-level implementation of BCI utilities provided as a main interface, offering a user-friendly experience.

### 2.1. OpenBCI: Fundamentals of Hardware and Software Components

OpenBCI is a highly flexible open-source hardware option for biosensing applications [30]. The board can work with EEG signals and supports Electromyography (EMG) and Electrocardiography. Furthermore, the biosensing board, as described in OpenBCI Cyton (https://openbci.com/, accessed on 1 October 2022), features a PIC32MX250F128B microcontroller, a ChipKIT UDB32-MX2-DIP bootloader, a LIS3DH 3-axis accelerometer, and an ADS1299 analog-to-digital converter with eight input channels (expandable to 16) with a maximum sampling rate of 16 kHz (as illustrated in Figure 1). The EEG channels can be configured in either monopolar or bipolar mode, with the capability to add up to five external digital inputs and three analog inputs. The Transmission Control Protocol (TCP) can also access data flow through a Wi-Fi interface. Table 3 summarizes the main configurations of OpenBCI. Another option is RFduino, which by default supports 250 Samples per Second (SPS) and eight channels. However, with the addition of the Daisy module, it can expand to 16 channels, and with the Wi-Fi shield, the sample rate can increase to 16 kHz. Of note, the whole electrode montage is configurable in monopolar, bipolar, or sequential modes. The Cyton board is equipped with Python-compatible drivers (https://github.com/openbci-archive/OpenBCI_Python, accessed on 1 October 2022), which have now been deprecated in favor of a new, board-agnostic family provided by BrainFlow (https://brainflow.org/, accessed on 1 October 2022). By developing board-specific drivers, the low-level features of the board will be integrated into their final version through high-level configurations. Computer-board communication is only occasionally reliable, and its GUI does not allow for data acquisition under specific parameters. However, the hardware and Software Development Kit (SDK) of the OpenBCI board offer the potential for implementing a complete framework comparable to medical-grade equipment [31].

### 2.2. High-Level Acquisition Drivers for OpenBCI

OpenBCI offers two main connection modes to communicate with the computer. The default connection is serial, where the computer recognizes the board as a serial device through a USB adapter that utilizes the proprietary RFDuino interface. It contains a Bluetooth-modified protocol designed to achieve high data transfer rates. With this interface, the maximum sample rate for eight channels is 250 Hz. Communication between the computer and the board is based on simple read-and-write commands, where the computer reads a specified number of bytes from the board or writes a specified number of bytes to the board. This streamlined communication protocol allows easy integration with various software applications. In addition to the serial interface, the OpenBCI system offers the option to increase the sample rate to 16k Hz with a Wi-Fi Shield. Yet, these high sample rates could be more practical for BCI applications, and it is easier to handle rates in the 1–2 kSPS. Further, the Wi-Fi connection supports Message Queue Telemetry Transport (MQTT) and TCP, with the latter being the preferred choice due to its simplicity. The OpenBCI is based on ADS1299, a 24-bit analog-to-digital converter from Texas Instruments designed specifically for biopotential measurements. It has been built using the ChipKIT development platform and its associated firmware. The Python-based SDK for OpenBCI defines an instruction set based on Unicode character exchange. This instruction set extends the system’s capabilities when operating in Wi-Fi mode. Besides, the protocols used for communication with the board are serial for the USB dongle and TCP for the Wi-Fi interface. Figure 2 illustrates the architecture of the proposed cross-platform drivers, starting with a foundation built on low-level features supported by OpenBCI’s SDK. The blue boxes in the figure represent these low-level features, which serve as essential building blocks for developing more complex functions. The SDK’s low-level features include tools and functionalities that provide access to raw BCI data and primary signal processing and feature extraction techniques. At the top layer of the drivers, the main interface, represented by the green boxes, provides access to a range of high-level elements essential for various BCI applications. These high-level features include data storage, marker synchronization, and impedance measurements. For example, the data storage feature ensures a secure and organized repository of large amounts of data, holding real-time synchronization of markers. Likewise, the impedance measurement feature is crucial for measuring the impedance between EEG electrodes and the scalp, ensuring the quality of signals. The proposed cross-platform drivers aim to provide a consistent and uniform interface across different platforms, making it easier for developers to write code once and run it on multiple platforms with minimal modifications.

–Select a block of binary data.–Prepend the offset data to the block.–Find the bytes header 0xa0 and slice the block with this byte as the first element and the remaining 33 bytes (at this point, the data is a list of arrays with a maximum length of 33 elements).–Crop the block of binary data to ensure the length of all elements is 33 and store the offset to complete the next block.–Create a matrix of shape (33,N). Now, the data structure on a shape (33,N) must meet a set of conditions: all first columns must contain a 0xa0 value; the second column must be incremental, and the last column must be in format 0xcX, holding the same value. Any row outside of these rules must be removed.

We perform data acquisition and deserialization separately to ensure a suitable structure. The module implements a multiprocessing queue for accessing data. The transmission format can be configured as RAW, binary, or formatted using JavaScript Object Notation (JSON), which has the advantage of already being deserialized, but the disadvantage of variable package size. Of note, RAW fixes the package size issue but requires a deserialization process. Still, RAW is fixed due to its fast transmission and ease of detecting lost packets. The primary EEG records are written in 24 bytes, compressing the eight channels of 24-bit signed data. This conversion is challenging for Python as there is no native 24-bit signed format. Therefore, a specific format has been implemented to interpret 16 channels using the same amount of data transmitted. Furthermore, Cyton and Daisy transmissions are interleaved, with empty blocks filled using the mean of the last two transmissions from the same board. The abovementioned process is described in Table 4. After acquiring the binary data, a deserialization process is necessary. This process consists of converting the bytes into values with physical units, i.e., μV for EEG and *g* for acceleration. Once the stream is started, a continuous flow of binary data is stored in a queue-based structure. Then, samples are processed to extract EEG and auxiliary information. Finally, a few steps must be implemented to deserialize the binary code package: Figure 3 shows graphically how a corrupted data set is cleaned and contextualized to deserialize the main structures. It is worth mentioning that the OpenBCI system allows for the customization of auxiliary data, which is set to accelerometer measurements by default. However, three signal types can be utilized: digital, analog, and marker. In marker mode, specific values can be programmed into the time series, while in digital and analog modes, signals can be inputted through physical ports. The ability to acquire external signals in addition to EEG signals is a crucial attribute, as it enables the system to measure latency values. After that, the presented high-level drivers leverage the OpenBCI configurations, as depicted in Table 5.

### 2.3. Distributed System for Fixed Latencies

The real-time capability of the introduced framework is defined based on the transmission of sampling blocks. Remarkably, our approach guarantees that EEG data blocks of duration *P* will be ready to use within a time less than *P*, regardless of the block’s duration [32]. This definition is necessary for comparing different system arrangements and assessing the flexibility in sampling rate, number of channels, protocols, and data block transmission. In addition, latency is expressed as a percentage to simplify the comparison of system capabilities when designing and developing a BCI. Further, the measurement process for latency, as shown in Figure 4, requires asynchronous measurements across multiple systems due to the distributed nature of the proposed interface.

The backbone of our distributed system is Apache Kafka, a robust platform for building real-time data pipelines and applications. Its ability to manage large amounts of high-throughput data, its simple protocol using a topic string identifier, and its capability to handle real-time data streams make it an ideal solution for many applications that require real-time processing and responsiveness. Hence, using a single server further accelerates message transmission by eliminating redundant message allocation and triggering. Furthermore, the versatility of Apache Kafka is enhanced by the availability of a Python wrapper called Kafka-Python (https://kafka-python.readthedocs.io/en/master/, accessed on 1 October 2022). As demonstrated by the real-time streaming of EEG data in a BCI system, Apache Kafka’s suitability for real-time processing and responsiveness is apparent.

For developing a distributed acquisition, a dedicated operating system only operating the essential processes and daemons (including the Apache Kafka server) is recommended using a Single Board Computer (SBC) with a minimal distribution of Linux, such as Archlinux ARM or Manjaro ARM Minimal. Upon boot, the system operates as a Real-Time Protocol (RTP) server and Wi-Fi access point, and the Apache Kafka server starts in the background. Next, the binary deserializer daemon begins listening for binary data, the EEG streamer listens for deserialized records, and the Remote Python Call (RPyC) server starts wrapping the drivers. RPyC is indispensable in this setup as it provides a transparent, symmetrical Python library for remote procedure calls, clustering, and distributed computing. It enables the execution of Python code on remote or local computers as if it were executed locally, making it convenient for both local and remote use. Moreover, RPyC enables the acquisition server to access the EEG efficiently while allowing the execution of Python code. Additionally, RPyC facilitates quick computation, even if the processing is performed remotely. The data can be effortlessly transmitted from the acquisition server to the processing server, guaranteeing a smooth flow throughout the system. In contrast to Apache Kafka, which handles large amounts of high-throughput data and provides real-time processing and responsiveness, RPyC establishes a connection between two Python processes and remotely access/control objects.

The overall architecture is depicted in Figure 5, showing the components and their interactions with other message protocols. The architecture includes three data transmission systems: PyC, Kafka, and Websockets. As highlighted in the yellow box, Kafka is crucial to maintaining a distributed communication system across the entire architecture. The green boxes represent other systems for tasks that do not require fast transmission or communication between terminals. The blue boxes represent all terminals executing specific tasks on independent computing units. If one terminal requires information from another, Kafka efficiently and reliably transmits the information, ensuring seamless communication between the terminals.

### 2.4. BCI Framework Interface

The proposed BCI framework is a powerful desktop application designed to provide a complete EEG-based BCI system on a single, user-friendly platform. The framework is developed entirely in Python and features a GUI built on PySide6, the latest stable release of the PySide library. Likewise, our use of open-source libraries and free software helps ensure the scalability and reconfigurability of the framework, allowing users to adapt the software to their specific needs quickly. The software is designed with a modular architecture, enabling almost all components to operate independently and communicate via Websockets or simple HTTP requests with the main interface. Furthermore, our strategy is optimized for use with the OpenBCI Cyton board, providing dedicated support for reliable EEG data acquisition. This optimization allows the main machine running the framework to allocate all its resources to data visualization, processing, and stimulus delivery, ensuring smooth data flow throughout the system. We use background services to run independent tasks, with some processes initiated from within the framework and others from outside. These services communicate using either Kafka or WebSockets, depending on the priority level or the information transmitted. This distributed network architecture helps ensure the efficient and reliable operation of the proposed BCI tool. Data analysis of the developed BCI framework is powered by Python and its extensive collection of scientific computing modules. The Python programming has proved suitable for developing neuroscience applications. Moreover, numerous modules, such as MNE (https://mne.tools/, accessed on 1 October 2022), are specifically designed to explore and analyze human neurophysiological data. The procedures in Numpy (https://numpy.org/, accessed on 1 October 2022) and Scipy (https://scipy.org/, accessed on 1 October 2022) can be used to implement custom analyses, and Scikit-learn (urlhttps://scikit-learn.org/, accessed on 1 October 2022) and TensorFlow (https://www.tensorflow.org/, accessed on 1 October 2022) used to implement machine and deep learning approaches.

Our framework allows building a BCI system by quickly accessing EEG signals and markers without worrying about acquisition, synchronization, or distribution. This approach allows real-time analysis to be implemented as a basic Kafka consumer or transformer that can connect to the EEG stream and consume the data to serve the user’s needs, generate reports, execute local commands, or send updated data back to the stream. The visualizations work similarly, except that they are limited to Kafka consumers as they are intended to be displayed within the BCI framework interface over HTTP instead of creating a new data stream. Real-time visualizations comprise a computational process that manipulates the data to create and update static visualizations. The visualization environment automatically serves the real-time EEG stream, allowing the user to focus solely on the visualizations. Concerning the interface for stimuli delivery, it is the only one that interacts directly with the BCI subject. Neurophysiological experiments require a controlled environment to decrease the artifacts in the signal and keep the subject focused on his/her task [33,34]. Consequently, the stimuli must be delivered over a remote presentation system, which physically separates the subject from the user. The method selected to develop the environment with these features is the classic web application, which is based on HTML, CSS, and JavasScript Brython-Radiant framework (https://radiant-framework.readthedocs.io/, accessed on 1 October 2022). Although this is a common feature in almost all neurophysiological experiments, after a series of observations and experience acquiring databases, we propose a brand new environment for designing, implementing, and configuring audio-visual stimuli delivery. Our interface allows the user to design flexible experiments and change the parameters quickly and easily without reprogramming the paradigm. Furthermore, since the acquisition interface is integrated into the framework, the database is automatically created with all the relevant metadata and synchronized markers. Then the user only has to worry about the experiment while the database is generated on a second plane.

Figure 6 outlines our OpenBCI-based framework, which integrates three key components into a unified system, offering a distinctive departure from conventional BCI systems where these components are often separate and poorly interconnected. The three components of our tool are the OpenBCI drivers, the distributed features, and the high-level interface. In a nutshell, our high-level interface provides a range of capabilities, including data analysis, real-time visualization, stimulus delivery, and an Integrated Development Environment (IDE). This integration results in a unified application interface with capabilities that can only be achieved through the synergistic relationship between the components. These capabilities include the creation of contextualized databases, real-time classifications, closed-loop implementations, the design of low-latency neurophysiological paradigms, and real-time visualization. Finally, multiple configurations and features of our framework hold online documentation at BCI Framework Documentation (https://docs.bciframework.org/, accessed on 1 October 2022).

## 3. Experimental Setup

We conduct a classical experiment in the BCI field to showcase the capabilities of the newly proposed OpenBCI-based framework, specifically a Motor Imagery (MI) paradigm. This experiment aims to demonstrate the benefits of our tool, including data acquisition, signal processing, and dynamic visualization. The ultimate goal of this test is to establish a robust database that will support the development of further offline processing stages, which can then be seamlessly integrated into the software for real-time feedback applications. Furthermore, by successfully testing the use of our framework in a well-established experiment, we aim to demonstrate its versatility, reliability, and usefulness in various BCI tasks.

### 3.1. Tested BCI Paradigm: Motor Imagery

MI is the act of imagining a motor action without actually performing it. For example, during an MI task, a participant visualizes in their mind a specific motor action, such as moving their right hand, without physically carrying it out. The planning and execution of movements result in the activation of characteristic rhythms in sensorimotor areas, such as α (8–12 Hz) and β (13–30 Hz) [35]. Investigating the brain dynamics associated with MI can have significant implications for various fields, including evaluating pathological conditions, rehabilitation of motor functions, and motor learning and performance [36]. As a result, BCIs that can decode MI-related patterns, usually captured through EEG signals, and translate them into commands to control external devices, have received much attention in the literature [35,37]. However, a significant limitation to the widespread adoption of these systems is that approximately 15–30% of users need help to gain control over the interface, as they do not exhibit specific task-related changes in sensorimotor rhythms during MI responses [38].

For concrete testing, our cue-based MI paradigm consisted of up to two different motor imagery tasks, represented by a sequence of cues (arrow-shaped) with asynchronous breaks. This paradigm utilized an arrow pointing to the left and right, which has been well established and widely applied in previous studies [39,40]. Figure 7 depicts the timeline of a single MI trial. It highlights the exact moment the system captures an event using markers, which will be integrated into the EEG signal time series for analysis. This trial is just one of many that will be administered to participants to build a comprehensive database of their brain activity during MI tasks using our OpenBCI-based approach.

### 3.2. Method Comparison and Quality Assessment

Most BCI experiments are highly dependent on acquisition quality. Indeed, the acquisition stage is crucial, as the system must perform with high accuracy, such as proper marker synchronization. During this stage, it is imperative to determine the precise moment when the participant was exposed to a stimulus by analyzing the EEG signal over time. Additionally, during the debugging phase of the experiment, several checks must be carried out to verify electrode placement and communication stability and validate the integrity of the collected data. These processes can pose challenges to the system. However, the easiest way to identify and resolve any issues is to visualize the data in real-time, in the time and frequency domains.

In addition, most approaches require multiple software tools, making it difficult to compare results. To accurately assess system performance, latency, and jitter are the most relevant measures [41,42]. Yet, due to varying sampling rates among acquisition systems, it is necessary to express these quantities as a percentage of the duration of the acquired data block. For example, if it takes 75 ms to transmit a 100 ms acquisition block, the latency would be expressed as 75%. Here, latency refers to the time elapsed between the raw EEG acquired from the board and its availability in the development framework. This analysis is performed on a fully distributed system with the following conditions:–OpenBCI acquisition system was housed within a dedicated SBC, specifically the Raspberry Pi was selected for its ease of use and the ease with which it could be transformed into a dedicated system [43,44].–Data are read in a remote computer using the developed drivers.–Block size is fixed at 100 samples according to latency analysis results.–Samples per second are fixed at 1000 due to latency analysis results.

Furthermore, the system is designed to register and stream at all times alongside the primary EEG data. This aspect enables developers to perform latency analysis without configuring a particular mode. This situation means that the same conditions used in an EEG acquisition session can be utilized for latency analysis. Figure 8 illustrates the experiment infrastructure implemented for this study. A Raspberry Pi card was configured as both an acquisition server and an access point, providing a direct connection to the OpenBCI acquisition system and reducing and stabilizing latency. The marker synchronization, real-time data visualization, and experimental paradigm configuration systems can be executed on any node connected to the network, either through a wired connection or wirelessly via the access point established by the Raspberry Pi. To minimize congestion in the acquisition data channel, all other nodes in this experiment were connected via a wired connection.

## 4. Results and Discussion

### 4.1. Impedance Measurement of EEG Electrodes

The impedance of the electrode-skin interface is an essential factor to consider in biopotential measurements, as it can significantly impact signal quality. Maintaining a low-impedance electrode skin is recommended to ensure low amplification levels, even below the resolution of the ADC. Our OpenBCI-based approach uses the ADS1299 ADC for biopotential measurements, which includes a method for measuring impedance using lead-off current sources. It involves injecting a small current of 6 nA at 31.2 Hz and processing the resulting signal to calculate the impedance using Ohm’s law. Nevertheless, the impedance measurement can be affected by nonstationary signals, such as during the placement or manipulation of the electrode. Therefore, it is recommended to allow for rest periods and follow best practices, such as taking short but sufficient signals and removing nonstationary segments, to improve impedance measurement accuracy. We can utilize the high-level driver to produce the results of a fundamental experiment where a 10 KOhm potentiometer is manipulated, depicted in Figure 9 that demonstrates how the measurement impedance fluctuates gently within the device’s range. Regarding actual skin-electrode impedance measurements, an acceptable range must be determined based on the electrode impedance and the ADC’s input impedance to reduce the amplitude variance caused by impedance and maintain comparable data channels.

### 4.2. Comparison of Results Based on Latency Analysis

Figure 10 compares four relative timestamps and the block duration for our OpenBCI-based tool within the tested MI paradigm. The binary time indicates the elapsed time from when the raw data was acquired to when it was streamed through Kafka. Likewise, the binary consumed time indicates the elapsed time from when the binary data was consumed to when it was deserialized. The produced time reflects the transmission duration, which is the time it takes for EEG data to be inserted into the Kafka stream until the final consumer reads it. Note that the difference between zero and EEG produced includes the clock offset. Therefore, the time between binary consumed and generated EEG includes the interval for deserializing the raw data. In contrast, the time between binary produced and Block duration represents the latency of the OpenBCI acquisition system when it operates over the WiFi protocol. Additionally, it is worth mentioning that efficient and accurate EEG data acquisition is crucial.

The process takes a long time, based on the results of deserialization. As seen in Figure 11, when the same process was performed for six different block sizes while maintaining the same 1000 SPS, the latency appeared to be linear for sizes under 1000 samples and up to 100. When represented as a percentage, the latency stabilized at around 50%. However, the greater the jitter, the longer the block size. This result suggests that the optimal configuration for EEG acquisition using the developed drivers is a block size of 100 samples with a jitter of only 8 ms. Figure 10 displays how latency dispersion, or jitter, increases as the acquired data passes through the different stages of transformation and transmission. This graph can be used to calculate the time between crucial tasks. The separation between the origin (0) and the red line corresponds to the EEG signal transmission time through Kafka. The distance between the red line and the orange line represents the deserialization time of the data as they are converted from binary to decimal. Furthermore, the distance between the orange and green lines represents the binary data’s propagation time, from when they are generated on the OpenBCI to when they are added to the Kafka stream, including transmission via the WiFi protocol. The blue line corresponds to the separation of each reading cycle in each iteration. For this experiment, the overall system latency can be calculated as the distance from the red line to the green line, approximately 56 ms.

Notably, our BCI stands out from traditional systems by performing experiments without needing a combination of systems or third-party applications. This limitation in comparisons between specific experiments, as highlighted in Table 6, underscores the importance of evaluating overall integrity rather than individual stages. Provided latency analysis results reveal the superior performance of wired systems in terms of lower jitter than wireless ones. Additionally, the latency of centralized implementations such as BCI2000 + g.USBamp can vary based on the paradigm used, leading to differing latency responses even with the same configuration. These findings highlight the need for careful consideration of the configuration and implementation of BCI to ensure optimal performance and desired outcomes. Nevertheless, our OpenBCI tool offers a suitable sample rate with an acceptable jitter level, even under a wireless implementation. Moreover, our approach maintains an acceptable latency for a low-cost, distributed, open-source framework, providing a cost-effective solution for EEG-based applications.

### 4.3. Sampling Analysis

In turn, for sampling analysis, a 64-minute continuous EEG signal is recorded at a sample rate of 250 SPS, with a block size of 100 samples and 16 channels. EEG channels are complemented with auxiliary data configured in signal test mode, which enables a square signal generator. To facilitate offline analysis, all channels are saved. In particular, to ensure the accuracy of the sampling rate, it is necessary to reject trials with glitches in the acquisitions caused by the transmission protocol. Three methods are employed to detect these issues: (i) analyzing the timestamp vector to detect any deviation in the step that indicates missing data; (ii) utilizing the square signal generator feature in the ADS1299 SDK to detect failed transmissions by observing variations in the pulse train in the auxiliary data; and (iii) examining the sample indexes, which consist of a looped incremental flag, to detect missing or duplicated data that will result in repeated or missing values in the sequence.

Figure 12 compares the three methods for detecting failed transmissions and shows their effectiveness. As depicted, all three methods effectively identify transmission failure points in the 64-minute signal. However, the sample index method is preferred due to its simplicity and efficiency. The missing samples are identified and marked as “BAD: sample” for further examination. Removing a section of samples surrounding each detection is recommended to account for the tendency of transmission failure points to cluster. The latter ensures that all trials containing one of these markers are excluded from the analysis. After identifying and removing flawed trials, the sample acquisition rate can be calculated. Figure 13 compares the sampling rate before and after eliminating markers designated as BAD samples. The left column shows the data without correction for sampling discrepancies, while the right column displays the data after discarding the identified BAD samples. The data in the right column is more refined and interpretable as it highlights the period difference between millisecond samples. The figure’s top plot represents the period difference, with a solid line marking the expected 4 ms (1/250 Hz) and a secondary line marking 50 ms equally distributed above and below. After removing the samples surrounding the “BAD: sample” markers, the bottom plot depicts the mean period for each resulting segment.

### 4.4. Interface Illustrative Capabilities

Our interface includes several features aimed at designing, developing, and debugging. Of note, it allows monitoring event marker synchronization within the tested paradigm. Synchronizing event markers in a distributed execution environment can lead to a slight delay in the EEG signal from when the stimulus was presented. This offset, which may not impact many BCI paradigms, can be problematic for such paradigms as Event-Related Potential and Motor Imagery, requiring differences of less than 10 ms. To address this issue, we propose synchronizing markers by implementing a Light-Dependent Resistor (LDR) module connected to pin D11 (or A5) of the OpenBCI and configured for analog mode acquisition. So, we calculate a single correction to the mean latency during the experiment, leveraging the low variability of latency designed into the framework. The main interface incorporates an automatic correction procedure that calculates and corrects this latency within the system. Corrections are made during the delivery of stimuli, with only a marker synchronization for the used area. The LDR module continuously senses changes in the square signal and compares them to the streamed markers. Figure 14 illustrates the calculation performed for a sequence of simulated markers to achieve a mean system latency of 0 ms. Furthermore, LDR can be applied per event, ensuring a precise marker located at the moment of each occurrence. It should be noted that the LDR module has to be continuously connected throughout the experiment. Figure 15 illustrates the implementation of this trial-based correction approach in a two-class MI recording, e.g., left vs. right. The last two plots demonstrate the contrast between global latency and trial-based correction.

Next, topographic visualization of EEG electrode impedance is presented in Figure 16, concerning our OpenBCI framework widgets. As seen, it facilitates cap tuning and checking by displaying the impedance value and the exact electrode and channel it belongs to. Similarly, real-time visualization of the acquired signals is an essential aspect of BCI experiments, as it enables the detection of any potential issues with the network connection or specific electrodes. The IDE in our BCI system provides powerful EEG visualization capabilities, both in the time and frequency domains, as demonstrated in Figure 17. Furthermore, with the IDE, custom visualizations can also be easily created to meet the specific needs of the experiment.

Furthermore, our IDE features a user-friendly Application Programming Interface (API), making it easy to create custom visualizations. Hence, users focus more on manipulating input data than worrying about acquisition parameters or signal transmission. For example, Figure 18 illustrates the composition of an introductory MI trial and demonstrates the pipeline system’s stability, ensuring the intervals between T0−T1, T1−T2, and T2−T3 remain consistent across all executions, despite fluctuations in view execution. Finally, Figure 19 shows the IDE interface used to create a simple motor imagery experiment. The environment includes an area for code editing, a preview of the designed interface, a file browser, and a debugging console. Once development is complete, this interface can be executed separately and serve remote stimuli to the patient through a specific IP that can be deployed through any browser.

## 5. Conclusions

We introduced a flexible, scalable, and integral OpenBCI framework for supporting EEG-based neurophysiological experiments. For such a purpose, the single-board OpenBCI Cyton was chosen, and a brand new set of drivers was developed to maximize the hardware benefits of the ADS1299. Our approach supports multiple sampling rates, packaging sizes, communication protocols, and free electrode placement, making it suitable for EEG data. Furthermore, an innovative feature for marker synchronization was also added. The system operates in a distributed manner, which allows for the controlled execution of critical processes such as acquisition, stimuli delivery, and real-time data analysis. Achieved results under a motor imagery paradigm demonstrate that the system’s robustness and stability are maintained through dedicated handling of the OpenBCI hardware. Real-time streaming is guaranteed within acceptable latency, and jitter ranges for closed-loop BCI compared to state-of-the-art approaches. The development environment provides a complete API, automatic background configuration, and a range of easy-to-use widgets for stimuli delivery, making it an ideal platform for BCI data processing and custom extension development.

Future work will extend the proposed framework to 32 and 64 EEG channels [47,48]. This work uses the OpenBCI Cyton board, one of the highest-performing hardware devices. However, we want to test upcoming acquisition boards integrating the most recent technology and communication protocols. Furthermore, close-loop approaches and advanced machine and deep learning algorithms will be tested to study poor skills issues with OpenBCI-based solutions [36,49].

## Figures and Tables

**Figure 1 sensors-23-03763-f001:**
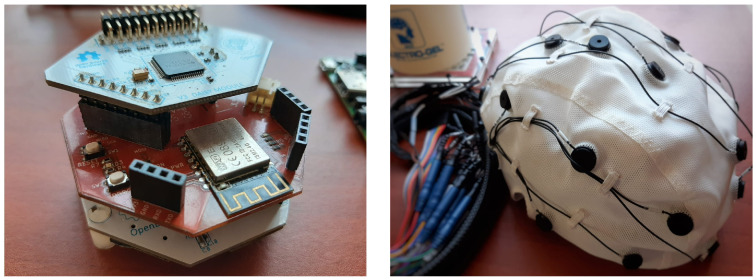
The OpenBCI system is equipped with a Daisy extension board and a WiFi shield for up to 16 channel support and 8 k SPS. It also includes an EEG cap with conductive gel for improved conductivity.

**Figure 2 sensors-23-03763-f002:**
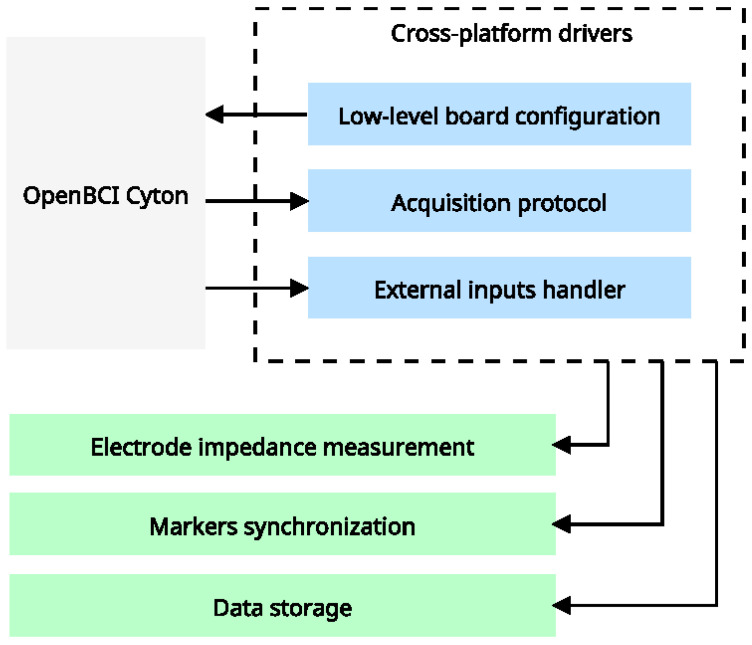
High-level acquisition drivers. Our architecture connects hardware features with the OpenBCI SDK to access all configuration modes, resulting in a cross-platform driver with low-level features (represented by blue boxes). External systems will utilize these features to deploy high-level, context-specific features.

**Figure 3 sensors-23-03763-f003:**
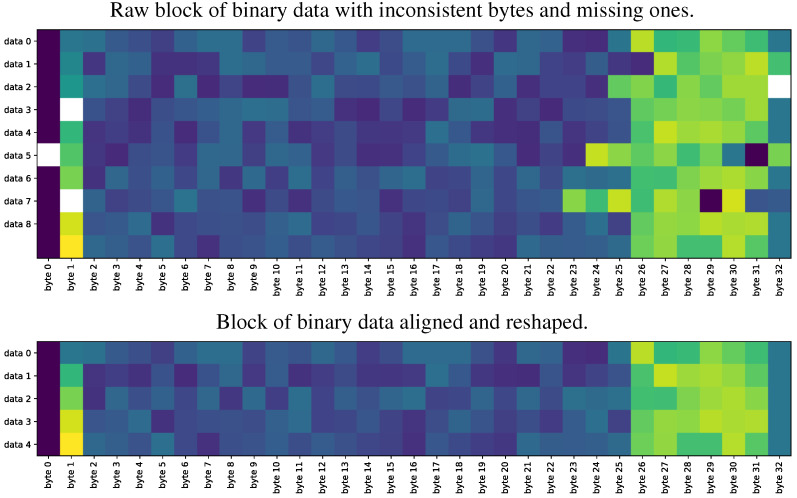
Proposed OpenBCI Cyton-based data block deserialization. Data deserialization must guarantee a data context to avoid overflow in subsequent conversion. The first block of columns, from 2 to 26, contains the EEG data, and the remaining ones, e.g., 26 to 32, gather the auxiliary data.

**Figure 4 sensors-23-03763-f004:**
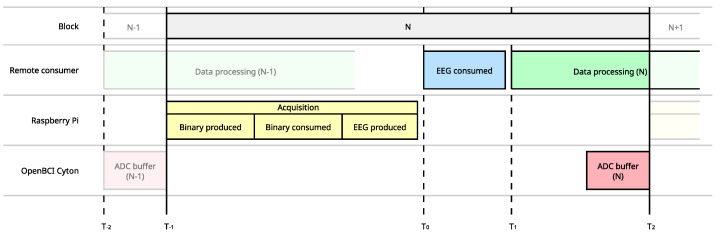
Composition of the data block throughout the distributed data acquisition and processing system,. A complete latency measurement must consider all systems where data is propagated. An accurate measure requires feedback and comparisons.

**Figure 5 sensors-23-03763-f005:**
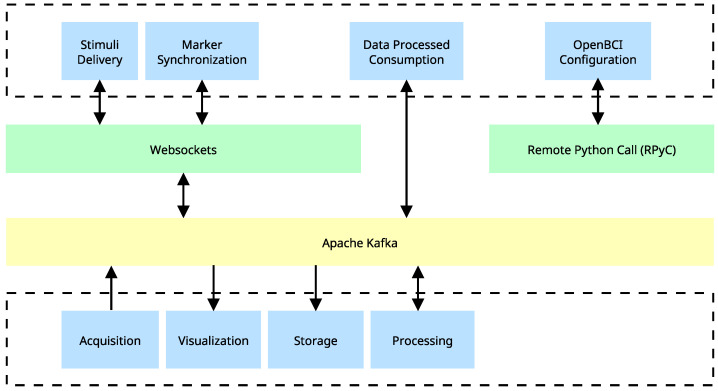
Distributed system implementation using Apache Kafka. Generation and consumption of real-time data are depicted, as well as the intermediation of a system that utilizes Websockets for additional tasks. Blue blocks depict tasks that can be executed independently across different processes or computing devices. The implementation of Apache Kafka serves as a mediator between various components and enables a seamless and efficient flow of information. Green blocks represent client-server communication protocols that are distinct from Apache Kafka.

**Figure 6 sensors-23-03763-f006:**
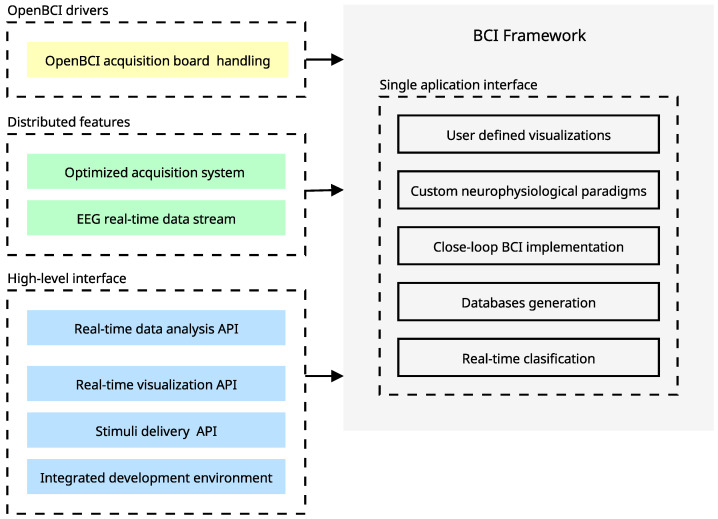
Proposed BCI tool based on OpenBCI (hardware/software) and EEG records. Our approach implements an end-to-end application. Beyond that, the synergy between this characteristic allows the achievement of advanced features that merge the data acquisition with the stimuli delivery in a flexible development environment. The yellow issue is related to the development of custom drivers for OpenBCI, the green ones refer to the integrations over distributed systems, and the blue issue refers to the high-level implementations of utilities served through the main interface.

**Figure 7 sensors-23-03763-f007:**
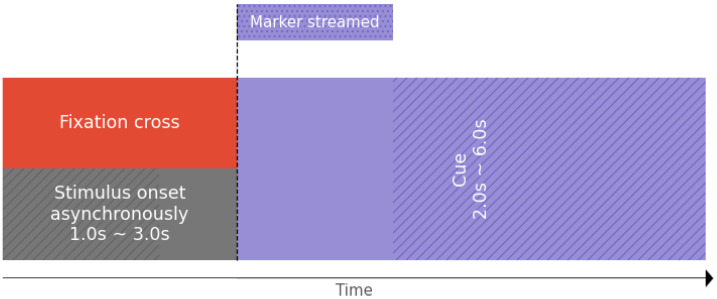
Tested Motor Imagery paradigm holding markers indicators for our OpenBCI-based framework.

**Figure 8 sensors-23-03763-f008:**
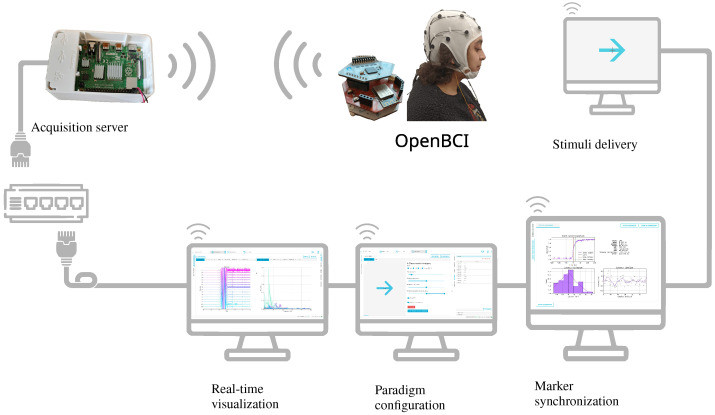
Implemented Motor Imagery experiment. Provided experimental setup utilizes a dedicated wireless channel for data acquisition and a wired connection for the distributed systems, enclosing real-time visualization, marker synchronization, paradigm configuration, and stimuli delivery.

**Figure 9 sensors-23-03763-f009:**
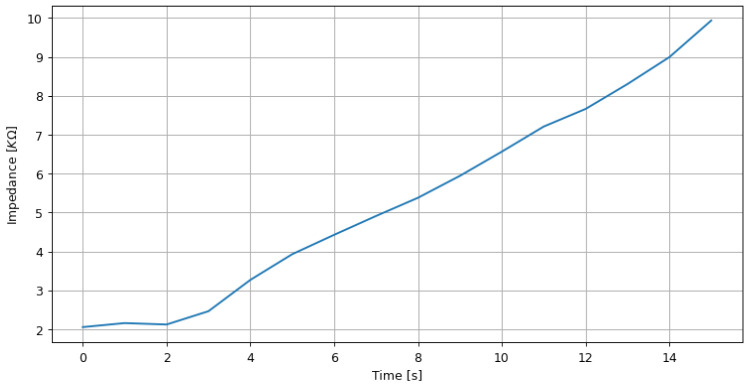
Real-time impedance measurement results by varying a 10 KOhm potentiometer.

**Figure 10 sensors-23-03763-f010:**
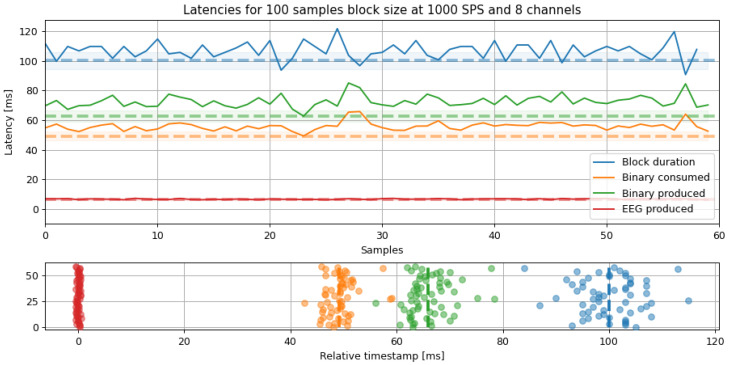
Latency analysis for a fixed 100 samples block size and 1000 SPS. The latencies show the elapsed time from reading the packet to the packaging time. The dashed line marks the minimum latency, and the shade is the standard deviation for all segments.

**Figure 11 sensors-23-03763-f011:**
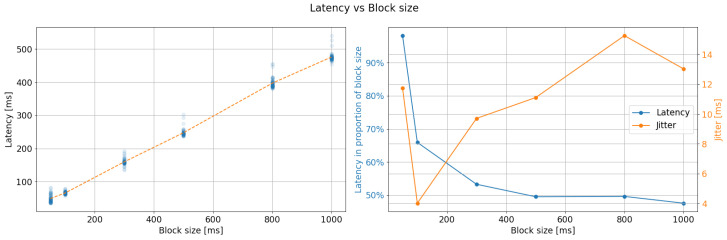
Latency vs. Block size results. **Left**: latency is proportional to the block size for small size values. **Right**: latency decreases for small block sizes but their standard deviation (jitter) increases for larger ones. The preferred configuration was set up in 100 samples block size due to the lower jitter.

**Figure 12 sensors-23-03763-f012:**
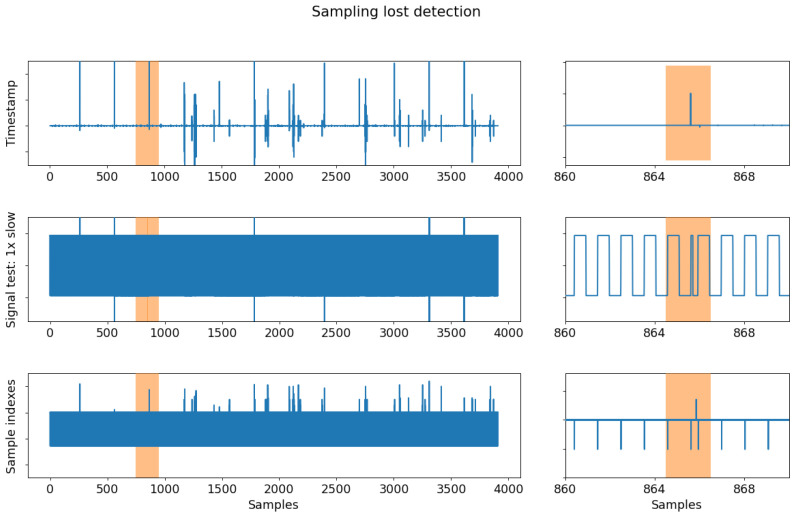
Sampling lost detection results. The sampling loss can be detected by analyzing the timestamp, using a test signal, or analyzing the sample index. Left column shows the complete time series in blue with a highlighted area. Right column presents a zoomed-in image regarding some samples irregularities.

**Figure 13 sensors-23-03763-f013:**
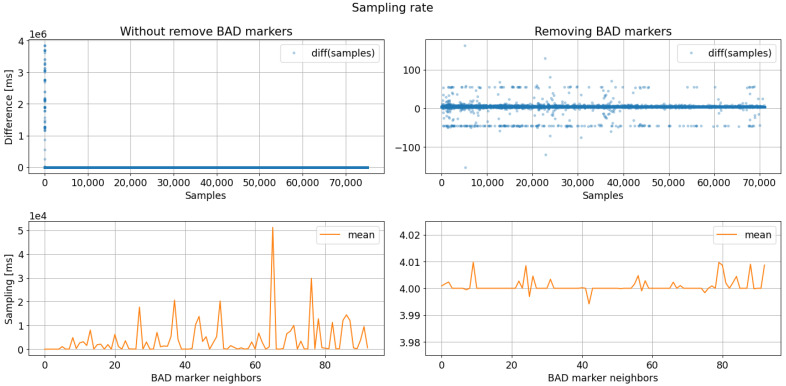
Sampling rate analysis results regarding bad markers removal. The sampling rate analysis after removing bad markers evidence a data period acquisition close to 4 ms (250 SPS).

**Figure 14 sensors-23-03763-f014:**
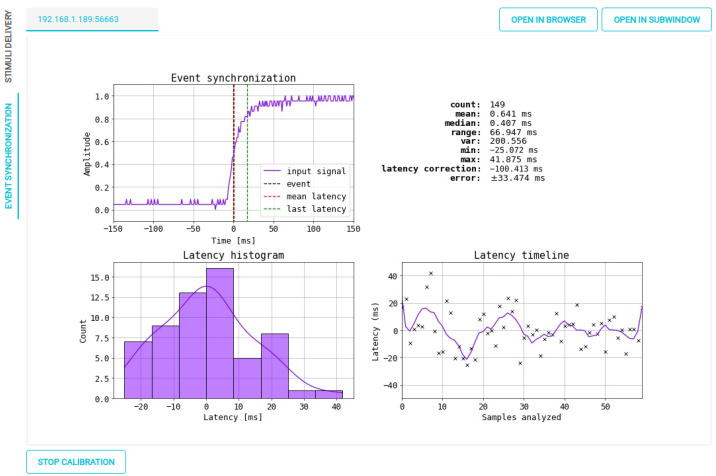
OpenBCI Framework results for marker synchronization within the real-time interface.

**Figure 15 sensors-23-03763-f015:**
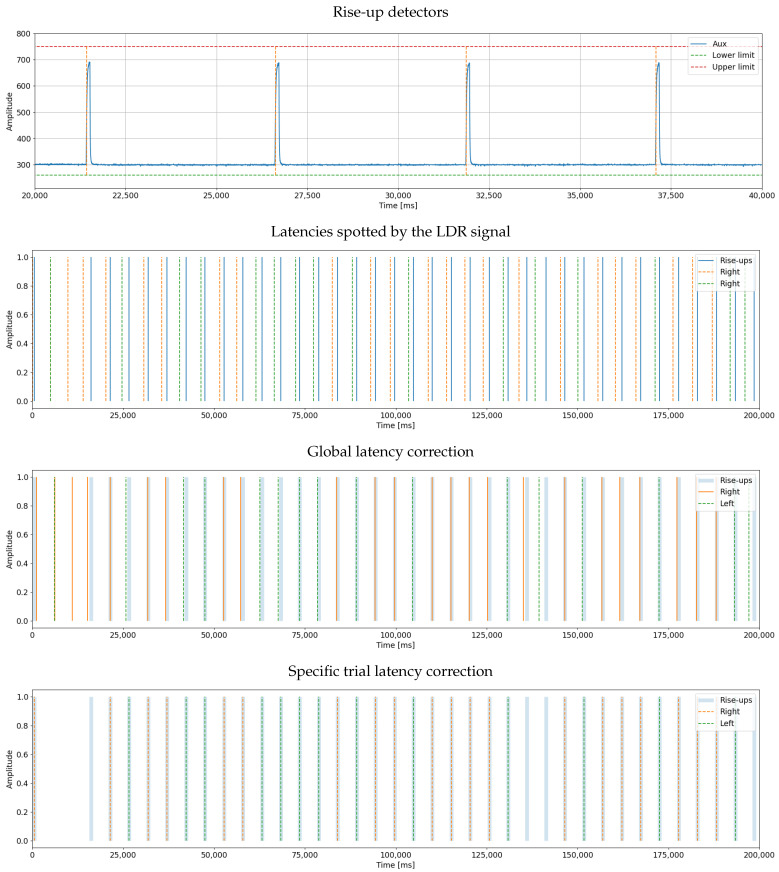
Automatic marker synchronization results. The top figure shows the signal generated by the LDR. The second figure visualizes the latencies in the system. The third figure shows what happens when all latencies are corrected with the same adjustment value, and the final plot has adjusted the latencies individually for each marker.

**Figure 16 sensors-23-03763-f016:**
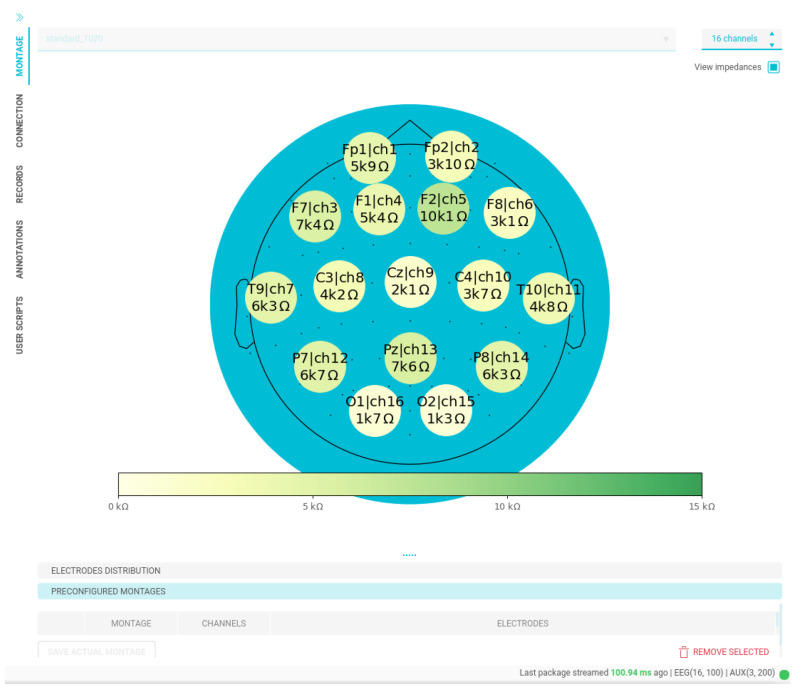
Interface widget for real-time impedance visualizer. OpenBCI EEG cap for 16 channels is used within our MI paradigm.

**Figure 17 sensors-23-03763-f017:**
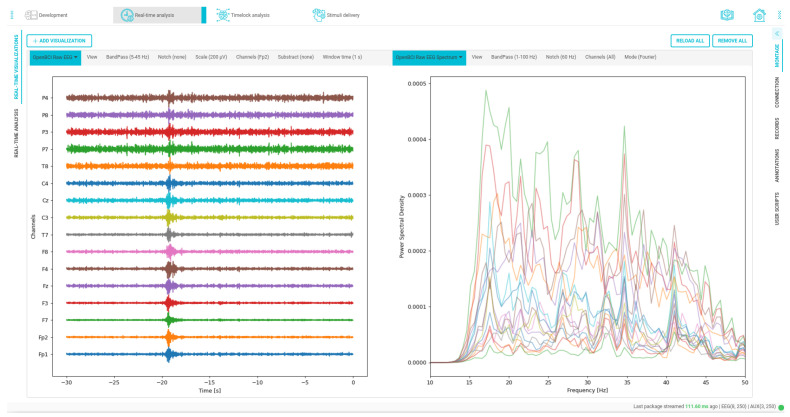
Time and frequency custom data visualizations. On the left side, the signal is filtered from 5 to 45 Hz, and on the right side, the spectrum of the channels is visualized within a bandwidth ranging from 1 to 100 Hz.

**Figure 18 sensors-23-03763-f018:**
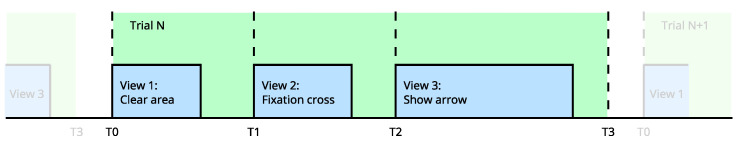
Stimuli delivery pipeline for MI paradigm. Each trial is composed of views; the pipeline features define the asynchronous execution of each view at the precise time.

**Figure 19 sensors-23-03763-f019:**
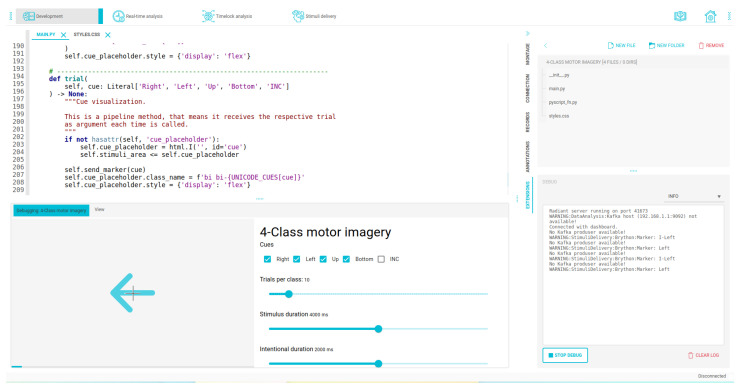
Integrated development interfaces displaying areas for code editing, file explorer, experiment preview, and debugging area for our OpenBCI Framework.

**Table 1 sensors-23-03763-t001:** Acquisition devices used for BCI. Bluetooth (BLE) devices tend to have slower speeds than Radiofrequency (RF) and Wi-Fi communication protocols. Wired-based data transfers, like USB, have the highest transfer rates.

BCI Hardware	Electrode Types	Channels	Protocol and Data Transfer	Sampling Rate	Open Hardware
Enobio (Neuroelectrics, Barcelona, Spain)	Flexible/Wet	8, 20, 32	BLE	250 Hz	No
q.DSI 10/20 (Quasar Devices, La Jolla, CA, USA)	Flexible/Dry	21	BLE	250 Hz–900 Hz	No
NeXus-32 (Mind Media B.V., Roermond, The Netherlands)	Flexible/Wet	21	BLE	2.048 kHz	No
IMEC EEG Headset (IMEC, Leuven, Belgium)	Rigid/Dry	8	BLE	-	No
Muse (InteraXon Inc., Toronto, ON, Canada)	Rigid/Dry	5	BLE	220 Hz	No
EPOC+ (Emotiv Inc., San Francisco, CA, USA)	Rigid/Wet	14	RF	128 Hz	No
CGX MOBILE (Cognionics Inc., San Diego, CA, USA)	Flexible/Dry	72, 128	BLE	500 Hz	No
ActiveTwo (Biosemi, Amsterdam, The Netherlands)	Flexible/Wet	256	USB	2 kHz–16 kHz	No
actiCAP slim/snap (Brain Products GmbH, Gilching, Germany)	Flexible/Wet/Dry	16	USB	2 kHz–20 kHz	No
Mind Wave (NeuroSky, Inc., San Jose, CA, USA)	Rigid/Dry	1	RF	250 Hz	No
Quick-20 (Cognionics Inc., San Diego, CA, USA)	Rigid/Dry	28	BLE	262 Hz	No
B-Alert x10 (Advanced Brain Monitoring, Inc., Carlsbad, CA, USA)	Rigid/Wet	9	BLE	256 Hz	No
Cyton OpenBCI (OpenBCI, Brooklyn, NY, USA)	Flexible/Wet/Dry	8, 16	RF/BLE/Wi-Fi	250 Hz–16 kHz	Yes

**Table 2 sensors-23-03763-t002:** BCI software. The most widely used software comprises free licenses: GNU General Public License (GPL), GNU Affero General Public License-version 3 (AGPL3), or MIT License (MIT). Usually, open-source tools admit extensibility by third-party developers.

BCI Software	Stimuli Delivery	Devices	Data Analysis	Close-Loop	Extensibility	License
BCI2000 (version 3.6, released in August 2020)	Yes	A large set	In software	Yes	yes	GPL
OpenViBE (version 3.3.1, released in November 2022)	Yes	A large set	In software	Yes	Yes	AGPL3
Neurobehavioral Systems Presentation (version 23.1, released in September 2022)	Yes	Has official list	In software	Yes	Yes	Proprietary
Psychology Software Tools, Inc. ePrime (version 3.0, released in September 2022)	Yes	Proprietary devices only	In software	No	Yes	Proprietary
EEGLAB (version 2022.1, released in August 2022)	No	Determined by Matlab	System Matlab	No	-	Proprietary
PsychoPy (vesion 2022.2.3, released in August 2022)	Yes	NO	NO	No	Yes	GPL
FieldTrip (version 20220827, released in August 2022)	No	NO	System Matlab	No	Yes	GPL
Millisecond Inquisit Lab (version 6.6.1, released in July 2022)	Yes	Serial and parallel devices	NO	No	No	Proprietary
Psychtoolbox-3 (version 3.0.18.12, released in August 2022)	Yes	Determined by Matlab and Octave	NO	No	-	MIT
OpenSesame (version 3.3.12, released in May 2022)	Yes	Determined by Python	System Python	No	Yes	GPL
NIMH MonkeyLogic (version 2.2.23, released in January 2022)	Yes	Determined by Matlab	NO	No	No	Proprietary
g.BCISYS	Yes	Proprietary devices only	System Matlab	No	No	Proprietary
OpenBCI GUI (version 5.1.0, released in May 2022)	No	Proprietary devices only	No	No	Yes	MIT

**Table 3 sensors-23-03763-t003:** OpenBCI Cyton configurations using Daisy expansion board and Wi-Fi shield.

OpenBCI Cyton	Channels	Digital Inputs	Analog Inputs	Max. Sample Rate	Featured Protocol
RFduino	8	5	3	250 Hz	Serial
RFduino + Daisy	16	5	3	250 Hz	Serial
RFduino + Wi-Fi shield	8	2	1	16 KHz	TCP (over Wi-Fi)
RFduino + Wi-Fi shield + Daisy	16	2	1	8 KHz	TCP (over Wi-Fi)

**Table 4 sensors-23-03763-t004:** EEG data package format for 16 channels.

Received		Upsampled Board Data	Upsampled Daisy Data
sample(3)		avg[sample(1), sample(3)]	sample(2)
	sample(4)	sample(3)	avg[sample(2), sample(4)]
sample(5)		avg[sample(3), sample(5)]	sample(4)
	sample(6)	sample(5)	avg[sample(4), sample(6)]
sample(7)		avg[sample(5), sample(7)]	sample(6)
	sample(8)	sample(7)	avg[sample(6), sample(8)]

**Table 5 sensors-23-03763-t005:** OpenBCI Cyton configurations using Daisy expansion board and a Wi-Fi shield.

OpenBCI Cyton	Channels	Digital Inputs	Analog Inputs	Max. Sample Rate	Featured Protocol
RFduino	8	5	3	250 Hz	Serial
RFduino + Daisy	16	5	3	250 Hz	Serial
RFduino + Wi-Fi shield	8	2	1	16 KHz	TCP (over Wi-Fi)
RFduino + Wi-Fi shield + Daisy	16	2	1	8 KHz	TCP (over Wi-Fi)

**Table 6 sensors-23-03763-t006:** Latency analysis for method comparison results. Latency has been expressed in terms of the percentage of the block size to make possible the comparison between different configurations.

BCI System	Sample Rate	Block Size	Jitter	Communication	Distributed	Latency
BCI2000 + DT3003 [45]	160 Hz	6.35 ms	0.67 ms	Wired	No	51.9%
BCI2000 + NI 6024E [45]	25 kHz	40 ms	0.75 ms	Wired	No	27.5%
BCI2000 + g.USBamp [32]	1200 Hz	83.3 ms	5.91 ms	Wired	No	14, 30, 48%
OpenViBE + TMSi Porti32 [46]	512 Hz	62.5 ms	3.07 ms	Optical MUX	No	100.4%
OpenBCI Framework (ours)	1000 Hz	100 ms	5.7 ms	Wireless	Yes	56

## Data Availability

The source code developed and the data employed for this research are publicly available at OpenBCI-Stream (https://github.com/UN-GCPDS/openbci-stream, accessed on 1 February 2022) and BCI framework (https://github.com/UN-GCPDS/BCI-framework, accessed on 1 February 2022) GitHub repositories. Moreover, detailed framework documentation is publicly available at BCI Framework Documentation (https://docs.bciframework.org/, accessed on 1 February 2022).

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
