# Peer review of "A Novel OpenBCI Framework for EEG-Based Neurophysiological Experiments"

_sensors, 2023, doi:10.3390/s23073763_

Round 1

Reviewer 1 Report

This paper proposes a novel OpenBCI framework for EEG-based neurophysiological experiments. The authors provided the detailed review on popular acquisition devices used for BCI systems, and then described the methods and software tools integrated in the proposed openBCI system. An example on motor imagery paradigm based on the developed openBCI system was used to illustrate its performance. Finally, some properties of the developed openBCI system were discussed.

From my point of view, this is a very interesting and complete work for promoting the EEG-based applications into practice. Nowadays, most of EEG-based BCIs research pay attention to EEG feature learning and classification models mainly while little work was put on system implementation.

The following comments are for the authors’ consideration.

1)     In the last paragraph of page 5, it mentioned ‘three components’. However, in this paragraph, there are ‘first’, ‘second’, ‘third’ and ‘finally’. Please check this point.

2)     The tables and figures I think should be arranged in order. For example, table 6 might be better put before figure 8.

3)     It is better to add a table to explicitly show the superiorities between the developed openBCI system and the existent one.

4)     Some figures (16 and 19 for example) should be enlarged to better view their fonts.

Reviewer 2 Report

- In the title, the two footnotes relating to the correspondence and equal contribution, were not included in the author list (e.g., there is not asteriskon the corresponding author name)

- The abstract in general need to be rewritten to be more crisp and to correct some style and typo mistakes (e.g., line 7 "operates distributed"). Also, the word "paradigm" is being misused. 

- Line 13, how did you measure and quantify robustness and stability?

- The introduction is too long and should be made more succinct and much shorter with clear motivation and contribution statements. 

- In general, each section of the manuscript can benefit from better flow and more organization. Currently, the paper is difficult to read/comprehend. Too many details are cluttereed all over the place. 

- Line 153, this is not an agenda, this an organization of what follows.

- The table of abbreviations is missing but required by the journal template. 

- Similar studies utilizing EEG and relating to brain activity and emotions should be discussed, which introduces practical applications to the work in the manuscript, see Fraiwan, M., Alafeef, M. & Almomani, F. Gauging human visual interest using multiscale entropy analysis of EEG signals. J Ambient Intell Human Comput 12, 2435–2447 (2021). https://doi.org/10.1007/s12652-020-02381-5

Round 2

Reviewer 2 Report

The authors addressed my comments.